# Impact of Field-Protective Forest Belts on the Microclimate of Agroforest Landscape in the Zone of Chestnut Soils of the Volgograd Region

Yustina Nikolaevna Potashkina and Alexander Valentinovich Koshelev *

Federal State Budget Scientific Institution "Federal Scientific Center of Agroecology, Complex Melioration and Protective Afforestation of the Russian Academy of Sciences", 400062 Volgograd, Russia
* Correspondence: koshelev_av@vfanc.ru

**Abstract:** Protective afforestation at the regional level is aimed at ensuring optimal agroecological conditions on agricultural land, including the regulation of microclimate on agricultural land. However, the issue of seasonal variability of microclimatic indicators in the zone of the forest shelterbelt and in different soil and climatic conditions is insufficiently studied. The research purpose is to determine the seasonal influence of aerodynamic parameters of the forest belt on the complex of microclimatic factors in the zone of chestnut soils of southern Russia. The research object is represented with agroforest landscapes of the Kachalinskoe experimental farm of the Ilovlinsky district of the Volgograd region. The study area is typical for this region in terms of soil cover and structure of protective forest plantations. The paper presents the results of study of the microclimate in the openwork-structure forest belt of a 30-year old. The ameliorative influence of the forest belt is determined by its height and construction. We have studied the microclimate indicators, such as wind speed, soil temperature and moisture, air temperature and humidity. We have carried out their instrumental measurements in the forest belt and in the adjacent territory at distances of 3H, 5H, 8H, 10H, 15H, 20H, 25H, 30H (H is the height of the stand in the forest belt) from the forest belt. The measurements were made once a season (summer, autumn, winter) in three-time intervals during the day. The research has shown that positive influence of forest belt on soil temperature in spring and autumn periods is traced up to 10–15H, in winter period there is no influence. The most distinct influence was noted in summer observations, so the average difference of temperatures in summer at 15H in the in the forest belt was 28%; in autumn, the same difference was 15%. The positive influence on soil moisture in the autumn period is most clearly traced, so the average difference in soil moisture indicators near the forest belt and agrolandscape is 18% and 2% in the summer period. The average temperature difference near the forest belt and 20H is only 4%, and 1% in the summer period. Effective reduction of wind speed occurs at 25–30H in the summer season, 15H in the autumn, and 15H in the winter periods. The difference in wind speed near the forest belt and average temperature in the agrolandscape is 52% in summer, 40% in fall, and 30% in winter. The results obtained are an attempt to assess the ameliorative impact of the forest belt on microclimatic indicators under conditions of regional climate change, and to make adjustments in the applied agricultural technologies for cultivation of crops in the inter-belt space.

**Keywords:** microclimate; protective forest plantations; wind speed; air temperature; soil temperature; soil moisture; agricultural landscape

## 1. Introduction

Climate change is recognized as a serious environmental problem of the 21st century. Climate change and unsustainable human activities lead to land degradation and desertification of territories. Protective afforestation is the most effective means of stabilizing the climatic situation in agrolandscapes in the fight against climate change [1–3].

In the regional context, protective afforestation is designed to solve the problems of regulating microclimatic parameters on agricultural land. The microclimate is understood as the meteorological regime of the lowest layer of air adjacent to the earth's surface, together with the near-surface soil layer. The microclimate plays a key role in crop production, since the growth and development of crops depends on the optimal combination of microclimatic parameters. Numerous studies in Russia and abroad have proven the direct impact of protective forest plantations on the microclimate. Starting with the works of the V.V. Dokuchaev's expeditions of in the Russian steppes (1892–1894) and the works of C.G. Bates in the USA (1911), the impact of forest belts on the microclimate of fields has been actively studied. It was found that protective forest plantations are able to reduce the wind flow velocity in adjacent fields [4–8], as well as air and soil temperature [9–14]. The microclimate created by forest belts expands the adaptive capabilities of cultivated plant species and varieties in the fields between belts [15–26]. Due to the reduced wind speed, increased air humidity, and reduced turbulent exchange, forest belts reduce evaporation from the soil surface, preserving soil moisture [27–30]. It was also proved that the microclimate was characterized by a significant spatial heterogeneity of the meteorological regime and a clear relationship with daily periodicity and with the influx of solar radiation [31–35].

Modern research into the precision farming have enhanced the study of the microclimate of agricultural landscapes to a new level. The use of high-resolution satellite images and unmanned aerial vehicles makes it possible to estimate the spatial distribution of microclimatic parameters and crop yields on a single field scale [36–39]. However, despite the variety of studies on this topic, there is still a lack of works focused on the spatial and temporal scale of studying the complex of microclimatic parameters. Of special interest are the studies carried out on the island of Hokkaido (Japan), which established the influence of protective forest belts and the conditions created by them (wind speed, temperature, and soil moisture) on the growth of corn [40]. We should also point to the study conducted in the arid zone of Northern China with the aim to assess the distribution on vertical (0–100 cm), horizontal (0–3H), and temporal (May–October) scales of soil properties, microclimate, soil moisture, and salt reserves during three growing seasons [41].

At the same time, the question of the variability of microclimatic parameters in a zone of protective forest belts depending on the year, season, and various soil and climatic conditions remains urgent. Therefore, the purpose of this work is to determine the influence of the parameters of the forest belt (construction, species composition, age) in different seasons on the complex of microclimatic factors (wind speed, soil temperature and moisture, air temperature and humidity) in the zone of chestnut soils in southern Russia.

The novelty of our research is due to its complex nature. We evaluated wind speed, air temperature and humidity, and soil temperature and moisture, depending on the construction of the plantation. The complexity of our research is also due to the different seasons of the study. At the time of our microclimate measurements the forest belt was in different states: with and without leaves.

## 2. Materials and Methods

The research object is represented with the agro-forestry landscapes of the Kachalinskoe experimental production farm located in the Ilovlinsky district of the Volgograd region (Figure 1). Coordinates of the test field center: 49°05′31.7″ N, 44°07′17.1″ E.

According to the agroclimatic zoning, the study area is located in a very warm arid region. The lowest air temperatures (−9.4 °C) are observed in January. July and August are the warmest months with average monthly temperatures of 23.2 °C and 21.5 °C, respectively. The average annual air temperature is 7.6 °C. The absolute maximum temperature reaches 43 °C in August, and the absolute minimum is −40 °C in January. The annual amount of precipitation is 385 mm. The largest amount of precipitation (up to 188 mm) is recorded in summer (June, July, August). In winter, the amount of precipitation is 87 mm, in autumn and spring it is up to 180 mm. The given research presents climatic characteristics for the last 30 years. The weather station is located in the village of Ilovlya of the Volgograd region.

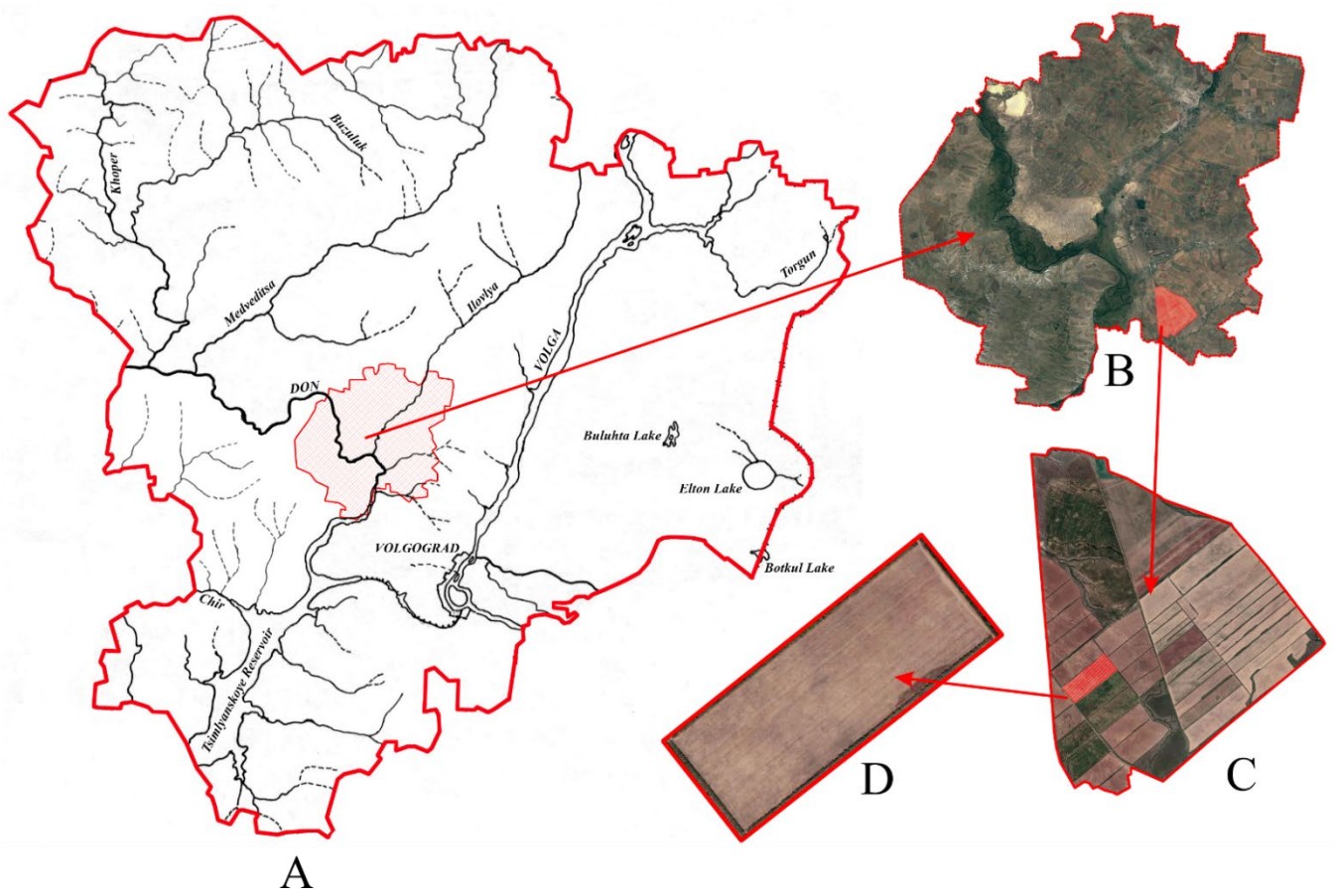

**Figure 1.** Schematic map of the research object location: (**A**) Volgograd region; (**B**) Ilovlinsky district; (**C**) Kachalino test site; (**D**) test field.

Most of the territory is located within the East-Don ridge on the hills, which are limited from the north, east, and south of the Don ridge. The territory is hilly, undulating, and flat, heavily indented by gullies and ravines. The dissection of the territory by ravines and gullies is about 3 km$^2$. The territory of the district is located in the zone of typical chestnut soils. There are also soil complexes with meadow subtypes of chestnut soils and chestnut solonets. Most of the natural chestnut soils and their complexes have been transformed into agricultural soils under the influence of agricultural activity. Various agro-chestnut soils and agrosolonets have become widespread in the area. At the object of the study, winter wheat Kamyshanka-4 (*Triticum aestivum*) is cultivated with the use of the classical system of bare fallow tillage.

When conducting a geo-information analysis of the investigated area, it was found that the terrain is leveled, and the slope did not exceed 1–2°. The field is protected from all sides by forest belts forming a square area. Based on the soil-geomorphological conditions, the area is representative and serves as a good example of the agrolandscapes of the Ilovlinsky district (Figure 2).

In our work, we used the methods generally accepted in the practice of agroforestry and agrometeorology [42]. Field studies were carried out in the summer, autumn and winter months of 2020–2021. The research was conducted according to the methodology of systematic studies of forest-agrarian landscapes (VNIALMI, 1985). All measurements of microclimatic parameters were carried out in 3-fold replications. In each ten-day period of the month, we selected 3 characteristic days and made the necessary meteorological measurements. As a result of the data obtained, we calculated the arithmetic mean values for all elements of the microclimate. The paper focuses on the analysis of changes in microclimatic indicators in agricultural landscapes depending on the distance from the

forest belt. We carried out a comprehensive measurement of microclimatic indicators, including the measurement of air temperature and relative air humidity, wind speed, soil temperature, and moisture. Wind speed, air, and soil temperature were measured using a CEM DT-620 480526 thermoanemometer. Relative air humidity was measured with a mechanical aspiration psychrometer MV-4-2M. Determination of soil moisture was carried out by thermostatic weight method with the use of the following equipment: auger, aluminum bucs, laboratory scales, laboratory balance, drying cabinet. Taxation surveys of the forest belt were carried out according to the methodology generally accepted in forest inventory and agroforestry [15,43–45]. The forest belt consists of 4 rows of trees, two of which are represented with Siberian elm (*Ulmus pumila*) and the other two—with marginal shrubs of golden currant (*R. aureum Pursh*). The age of the forest belt is 30 years.

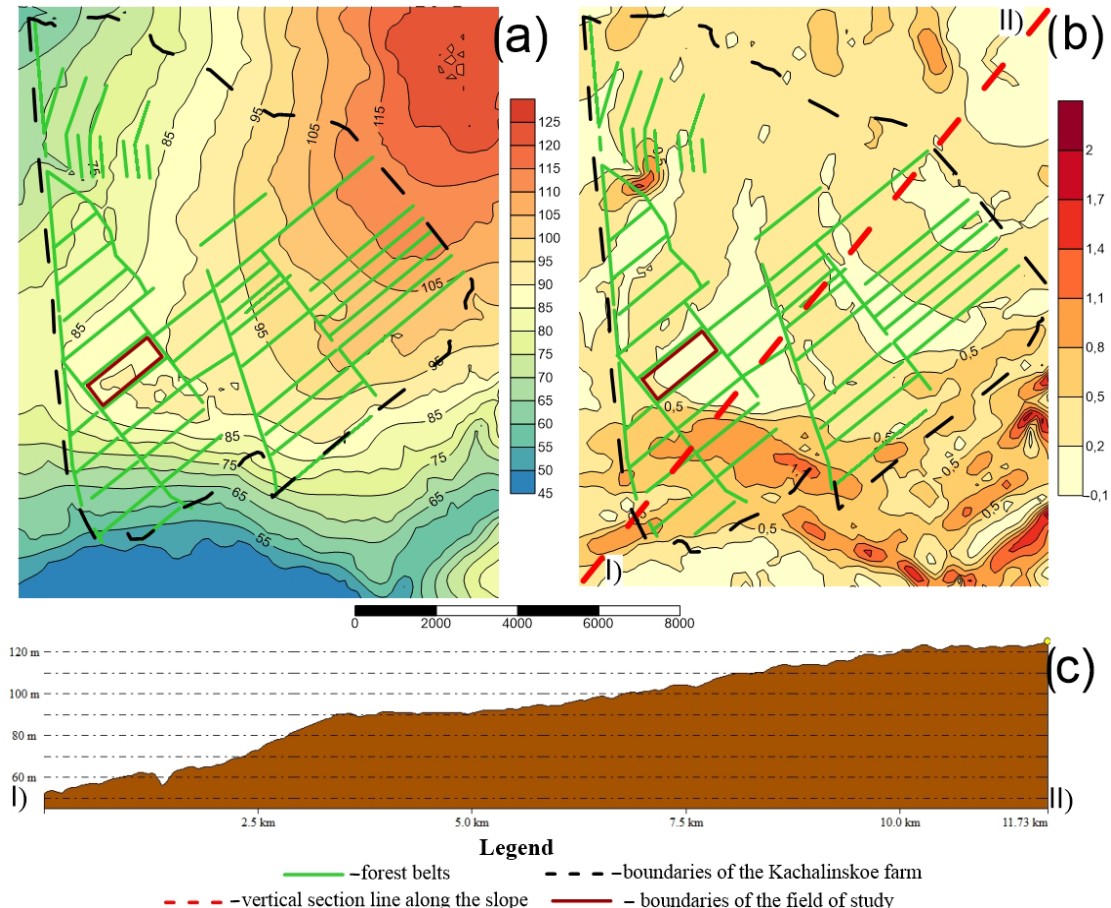

**Figure 2.** Maps of relief (m) (**a**), slope steepness (degrees) (**b**), and vertical section along the slope (**c**) of the investigated area.

The length of the forest belt is 1300 m, the width is 12 m. The forest stand is highly preserved in the belt. The average height (H) of trees is 10 m, the average diameter of tree trunks is 17 cm. According to the scale of quality classes, the forest belt belongs to the 2nd class. In general, the state of the forest belt can be assessed as satisfactory. Only 5% of the total number of stands are dry-topped [46] (Figure 3).

When determining the degree of openwork of the forest belt, we used the authors' method based on the classification according to the maximum likelihood method using the ENVI software (Exelis Visual Information Solutions, Boulder, CO, USA). At the same time, 3 classes were chosen as the basis: 1—the green (deciduous) part of the tree plantation; 2—dark and brown tones of the trunks; 3—light colors of the sky. The accuracy of the classification can be judged by the value of Kappa Coefficient: the higher this indicator, the more accurate the classification of a photo image. Kappa Coefficient made 0.9939 (99%

accuracy) when classifying the photos of the forest belt. According to the result of the analysis, it was found that the gaps accounted for 21.3%, and the impermeable and woody parts—78.7%, respectively. According to E.S. Pavlovsky's classification, the belt has an openwork construction [42].

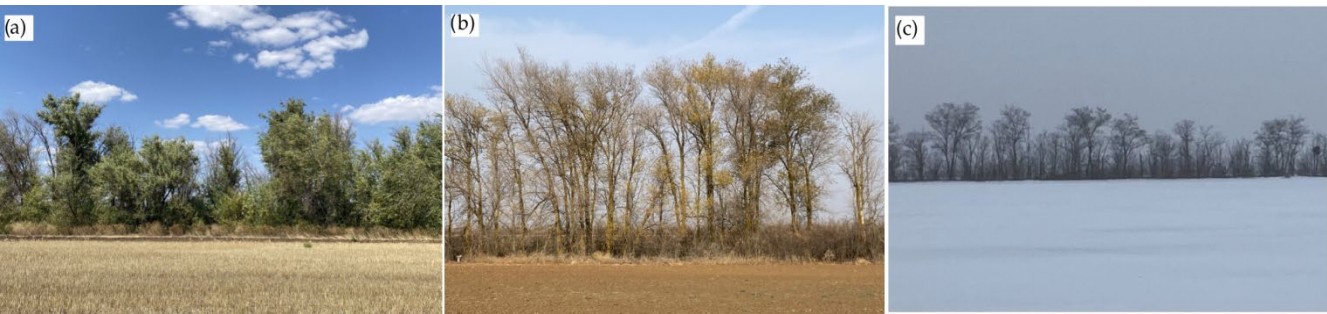

**Figure 3.** The forest belt during the periods of research: (**a**) summer 2020, (**b**) autumn 2020, (**c**) winter 2021.

As part of the research, we conducted an experiment that consisted of determining the following microclimatic indicators: wind speed, air and soil temperature, relative air humidity, soil moisture. Measurements of wind speed, air, and soil temperatures, as well as relative air humidity were carried out directly in the forest belt and at the following distances from the forest belt—1H, 3H, 5H, 8H, 10H, 15H, 20H, 25H, 30H. All microclimatic indicators were measured three times at two heights of 0.5 and 1.5 m from the earth's surface 3 times a day (10 a.m., 1 p.m., 4 p.m.). Soil moisture was determined in the forest belt at the distances of 1H, 5H, 10H, 20H, 30H, and at the depth of up to 50 cm, every 10 cm (Figure 4) [46–48].

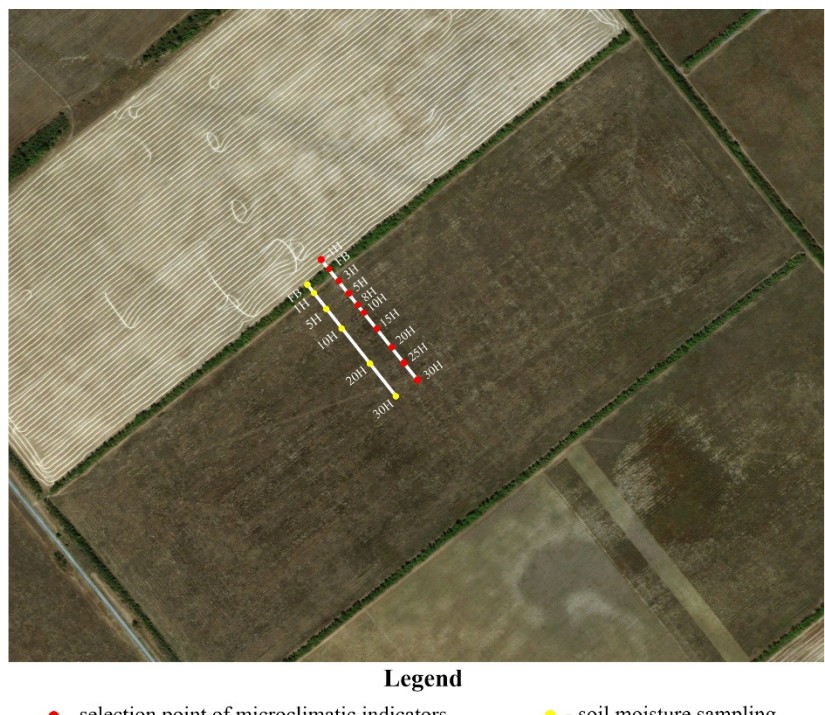

**Legend**

● - selection point of microclimatic indicators          ● - soil moisture sampling

**Figure 4.** Soil sampling and measurement points in the investigated area.

In the agroforestry studies, it is optimal to study the layer of 0–100 cm when establishing the role of forest belts in changing soil moisture. However, when establishing the type and subtype of soils in the studied agrolandscape, we found that, starting from a depth

of 50 cm and deeper, there is a horizon of accumulation of secondary carbonates, which hinders the infiltration of sediments due to a heavier soil particle size distribution. In this regard, we decided to analyze the upper 50 cm of the soil [45,46,49].

## 3. Results

### 3.1. Soil Temperature

In the morning (10 a.m.), the most distinct impact of the forest belt is observed in summer. In the leeward side, the soil temperature is equal to 51°C, and in the forest belt—27.8 °C. The forest belt has the most active cooling effect in the morning up to a distance of 3H (30 m) from the forest belt. Thus, the minimum value of soil temperature in the agrolandscape (40 °C) is observed at the distance of 3H. At a distance of 3–30H, sharp temperature changes were not observed. In autumn, in the leeward side in the morning, the temperature is lower than in the rest of the agrolandscape. In autumn and winter, the impact of the forest belt is almost absent. Height (H) of the forest belt 10 m.

In the afternoon (1 p.m.), the soil is warmer; the average soil temperature is 4.8 °C higher than in the morning. The cooling effect of the forest belt at 1 p.m. in summer can be traced on average up to 10H, then the temperature is stable. In autumn, a distinct zone of the belt's impact stretches up to 5H, while in winter, it is completely absent.

In the evening (4 p.m.) in summer and autumn, the soil temperature has the highest values in the leeward side at a distance of 1H—43.6 °C and 28.6 °C, respectively. At a distance of up to 10H, the temperature gradually increases, reaching maximum values of 46.6 °C in summer and 26.6 °C in autumn. At a distance from 10 to 30H, the temperature gradually decreases from 46.6 °C to 39.6 °C in summer and 26 °C to 21.3 °C in autumn (Figure 5).

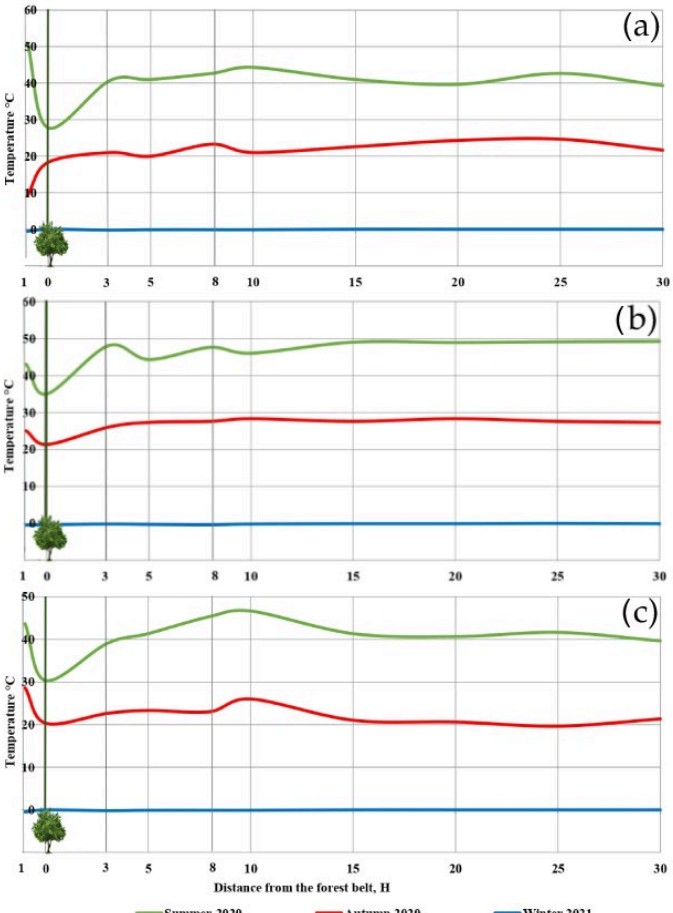

**Figure 5.** Soil temperature in (**a**) the morning, (**b**) the afternoon, and (**c**) the evening.

### 3.2. Soil Moisture

In the summer season, the highest values of moisture (up to 12%) of the upper 0–20 cm soil layer are recorded in the forest belt due to the fact that it contains a thick layer of leaf litter. Moreover, the soil is in the shade for most of the daylight hours, and, therefore, evaporation is lower. Further from the forest belt, at a distance of 1H, the lowest values of soil moisture are recorded. Thus, soil moisture in the 0–30 cm layer is 6.3%, and in the 0–50 cm layer 7.31%. First of all, this is due to the fact that the forest belt has an openwork design, and consequently, the wind speed is much higher at a distance of 1H, and additional drying of the soil profile occurs. At a distance of 5H, we have identified a "depression zone". The depression zone is the area of the agrolandscape closest to the forest belt, which experiences the greatest impact of the forest plantation. In this zone, high values of moisture are recorded, both in individual layers and in the entire 50 cm thick layer. Moisture in the 0–30 cm layer is 8.3%, and in the 0–50 cm layer 10.5%. At a distance of 10H, the humidity in the 0–30 cm layer is 1% lower than at a distance of 5H and is equal in the 0–50 cm layer (10.4%). Further, the territory of the agrolandscape represents an alternation of both absolutely flat surface areas and micro-hollows. Soil moisture in the micro-hollows is higher than on an absolutely flat surface. For example, the moisture at 20H in the 0–30 cm layer is 9.5%, and in the 0–50 cm layer—12.0%. At the control distance from the forest belt (30H), the moisture is approximately equal to the moisture in the 10H zone. Moisture of the uppermost layer (0 cm) in the entire agricultural landscape is at the level of 3.0%. Below this level, the moisture values may vary depending on the slope and exposure, but no significant differences were found (Figure 6).

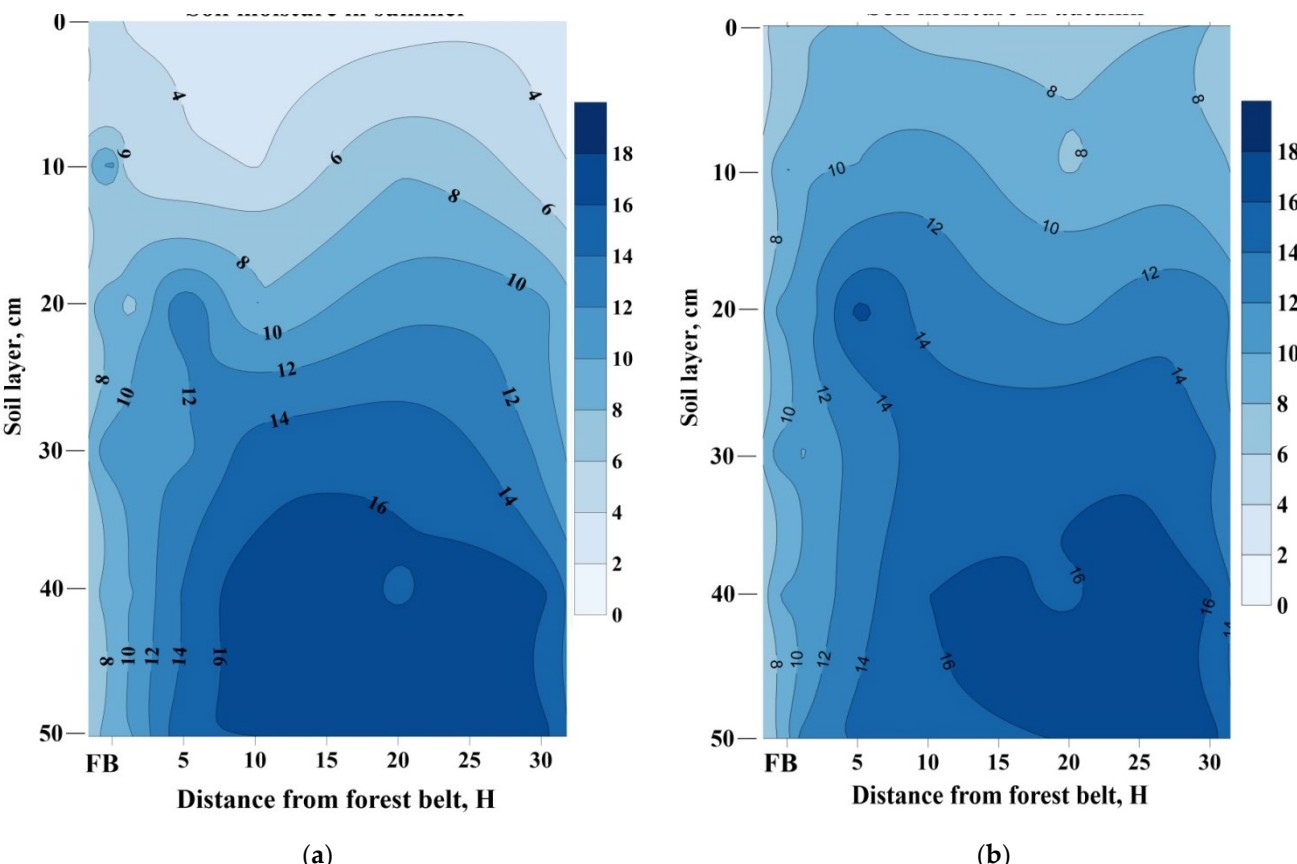

**Figure 6.** Soil moisture in the Kachalinskoe farm: (**a**) summer 2020, (**b**) autumn 2020.

The nature of soil moisture in the autumn season is noticeably different from the summer one. Due to the decrease in air and soil temperatures, evaporation from the surface noticeably decreases. So, if the humidity on the surface in the summer period averages

3.0% in the agricultural landscape as a whole, then in autumn, it is at the level of 8.0%. Due to the autumn rains, wetting of the soil profile is at the level of 100 cm in the forest belt and 80 cm in the field part of the agricultural landscape. In summer, soil moisture in the forest belt in the upper 0–20 cm level is noticeably higher than in the field, and in autumn, due to atmospheric precipitation, the moisture content of this layer is almost the same both in the plantation and in the field. This is primarily due to the moisture uptake by tree roots, as well as the physical barrier between rainwater and the soil surface. As in summer, the minimum values of humidity are recorded in 1H zone. However, in summer this takes place in all layers, while in autumn the minimum values are observed in the layers ranging from 20 to 45 cm. The soil layers of 0–15 cm at a distance of 20H and in the control area (30H) turn out to be the least moistened. The maximum values of soil moisture are recorded at a depth of 40–50 cm, since it is from these layers that the carbonate horizon begins. It is heavier than the overlying ones in terms of particle size distribution and makes the infiltration of sediments more difficult.

In general, both in autumn and in summer, there is no direct dependence of soil moisture on the distance from the forest belt, since at different distances and in different layers, the humidity can be higher or lower. The differences do not exceed 2.0% on average. At a distance of 5–10H, a depression zone was identified, since the moisture values in the layers were higher than in the layers at other distances. The degree of moisture in the autumn period is uniform, while in summer it is characterized by moisture accumulation zones (lower and middle layers) and moisture deficit zones (upper 0–15 cm layer).

*3.3. Air Temperature*

In the morning hours in summer, the temperature varies from 0.1 °C to 0.5 °C at the height of 0.5 m and 1.5 m, respectively. The highest values of air temperature are recorded in the leeward side: 31.7 °C and 32.4 °C, respectively. There are no sharp changes in the dynamics of air temperature depending on the distance from the forest belt. The air temperature curve is linear. The impact of the forest belt on the air temperature is weakly expressed. Thus, the temperature difference between 3H and 30H is only 0.6 °C and 0.4 °C, respectively. The maximum value of air temperature (29.8 °C) is noted at a distance of 20H (Figure 7).

In the autumn and winter periods, no sharp changes in air temperature are observed at both altitudes at 10 a.m. This allows us to assert that the influence of the forest belt is weakly expressed. On the leeward side, the highest air temperatures are observed 5 °C and 4.6 °C, respectively. In summer, there are no sharp changes in air temperature at the altitudes of 0.5 m and 1.5 m in the afternoon. Thus, the air temperature at a distance of 5H is 28.3 °C and 28.5 °C, respectively, at 30H 28.7 °C and 28.8 °C, respectively. Based on this, it is difficult to single out the zone of impact of the forest belt. In winter, the warming effect is observed at 1 p.m. at the heights of 0.5 m and 1.5 m in the forest belt itself. Further, at all measurement points, the temperature difference does not exceed 0.5 °C.

In summer evenings, the temperature values at the given altitudes almost completely coincide. The differences are noted only at a distance of 30H: 34.4 °C at 0.5 m, and 33.6 °C at 1.5 m. It can also be argued that there is no pronounced effect of the forest belt on air temperature. In winter, evening measurements are characterized by similar results. A warming effect is observed at a distance of up to 3H from the forest belt.

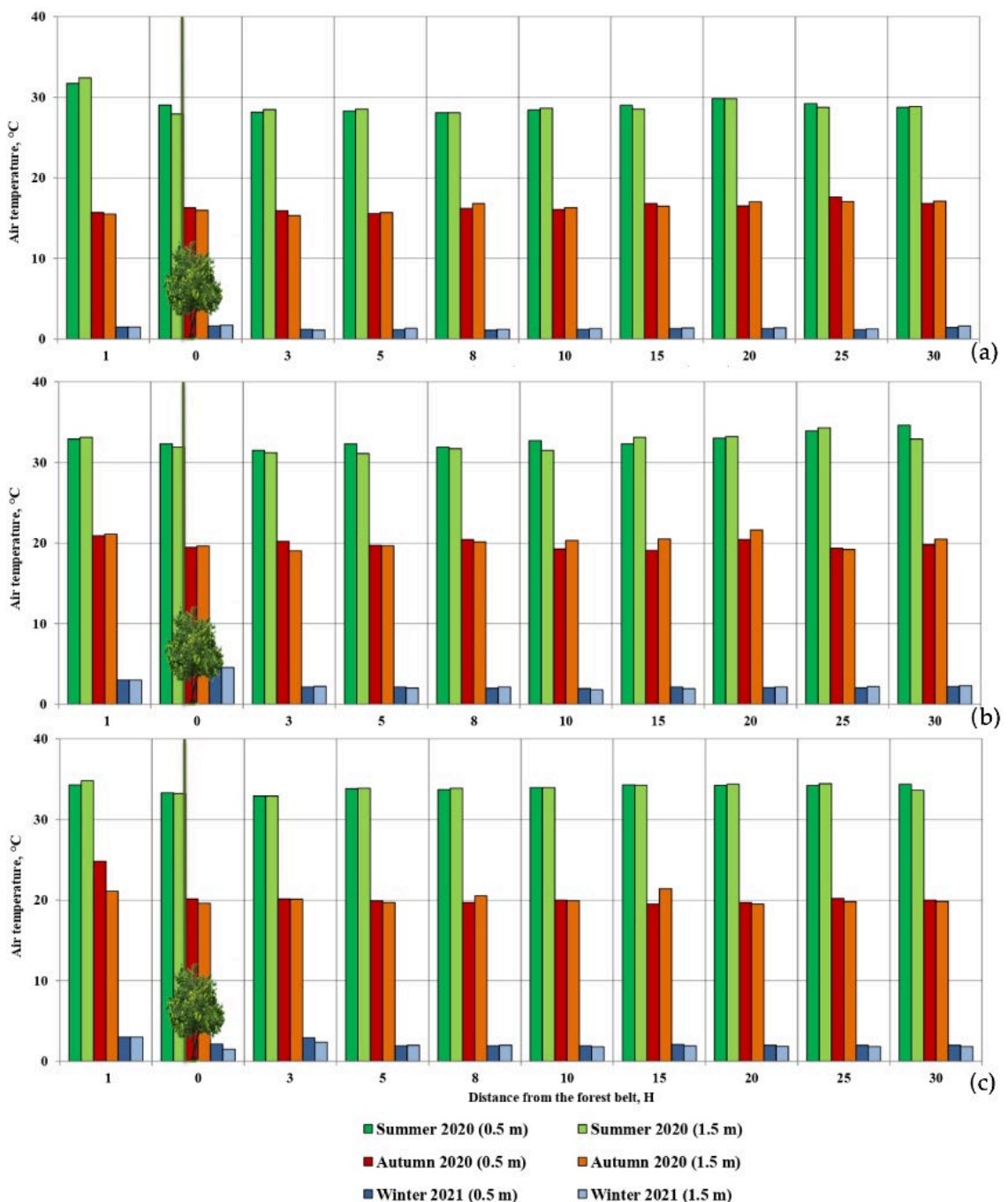

**Figure 7.** Air temperature in the agrolandscapes of the Kachalinskoe farm in 2020–2021: (**a**) 10 a.m., (**b**) 1 p.m., (**c**) 4 p.m.

### 3.4. Air Humidity

When measuring the humidity of the air in the morning hours in summer, no sharp differences are noted in the field part. The difference at the distances of 3H and 30H is 3% at both altitudes 0.5 m and 1.5 m. The maximum value of relative air humidity is recorded in the forest belt: 61% at 0.5 m, and 63% at 1.5 m. In the autumn period, the difference between 3H and 30H in the morning hours is also less than 3% at both altitudes. The difference between morning measurements in winter is also not large and amounts to 2% in the field part (Figure 8).

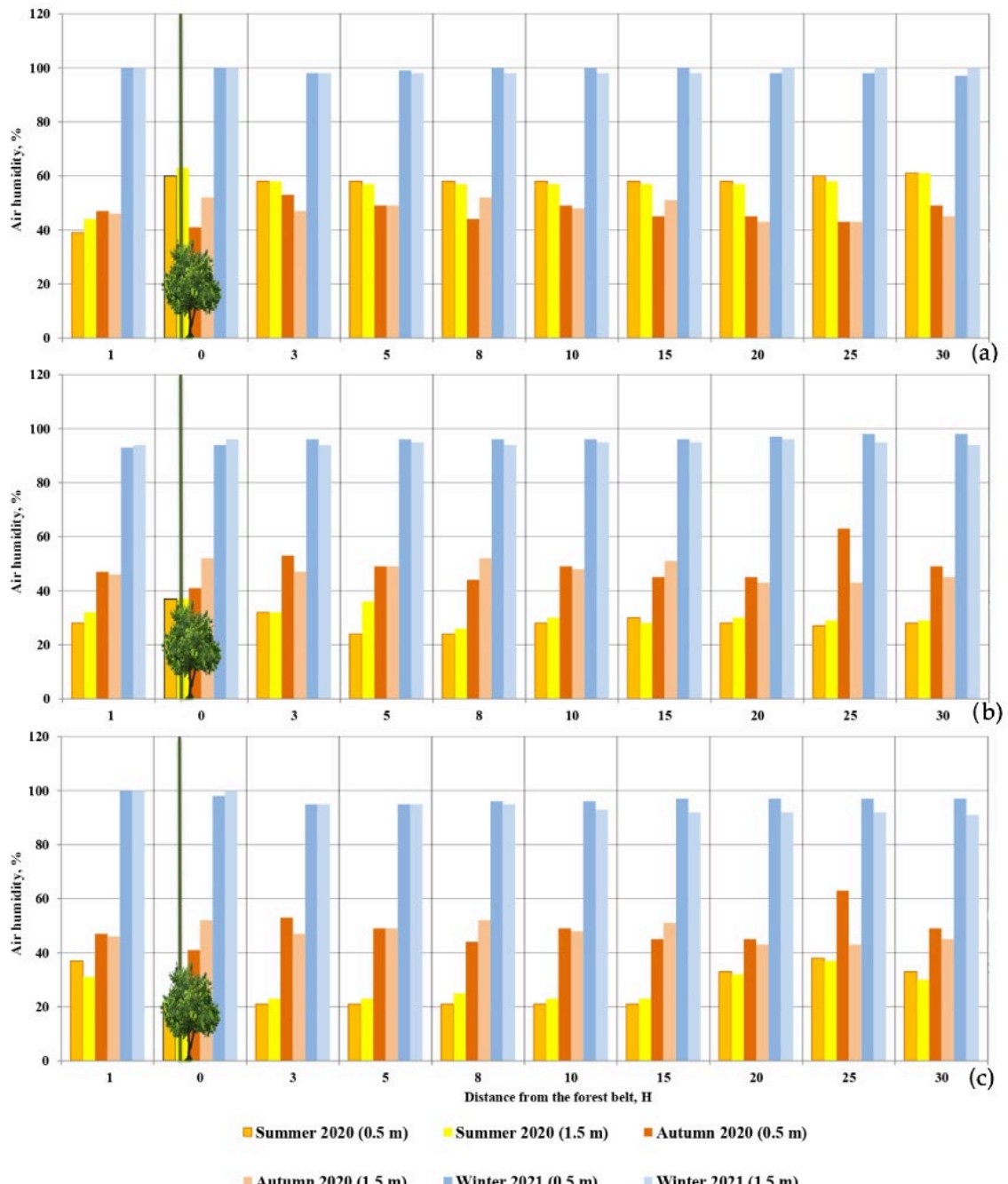

**Figure 8.** Air humidity in the agrolandscapes of the Kachalinskoe farm in 2020–2021: (**a**) 10 a.m., (**b**) 1 p.m., (**c**) 4 p.m.

In the afternoon, the average indicators of air humidity are higher than in the morning due to higher air temperatures. The difference between values at 3H and 30H at two altitudes in summer and autumn increases by 3%–5%, and in winter it decreases by 2% after 4:00 p.m.

### 3.5. Wind Speed

In summer mornings, at the given altitudes, the "classical" impact of the forest belt is recorded. Thus, the wind speed increases in direct proportion to the distance from the forest belt. We did not find any differences between the indicators at 0.5 m and 1.5 m. In summer, the range of the forest belt impact is traced up to a distance of 20H. In autumn, at a height of 0.5 m, the wind speed gradually increases to a distance of 5H (2.4 m/s) and

then does not change to 30H, where the maximum value of 3.3 m/s is recorded. At a height of 1.5 m, a similar situation is observed. The difference in wind speed values at 0.5 m and 1.5 m is only 0.2 m/s. In winter, the wind speed curve at 0.5 m and 1.5 m is linear, without sharp changes. The average wind speed is 5.5 m/s at 0.5 m, and 5.2 m/s at 1.5 m (Figure 9).

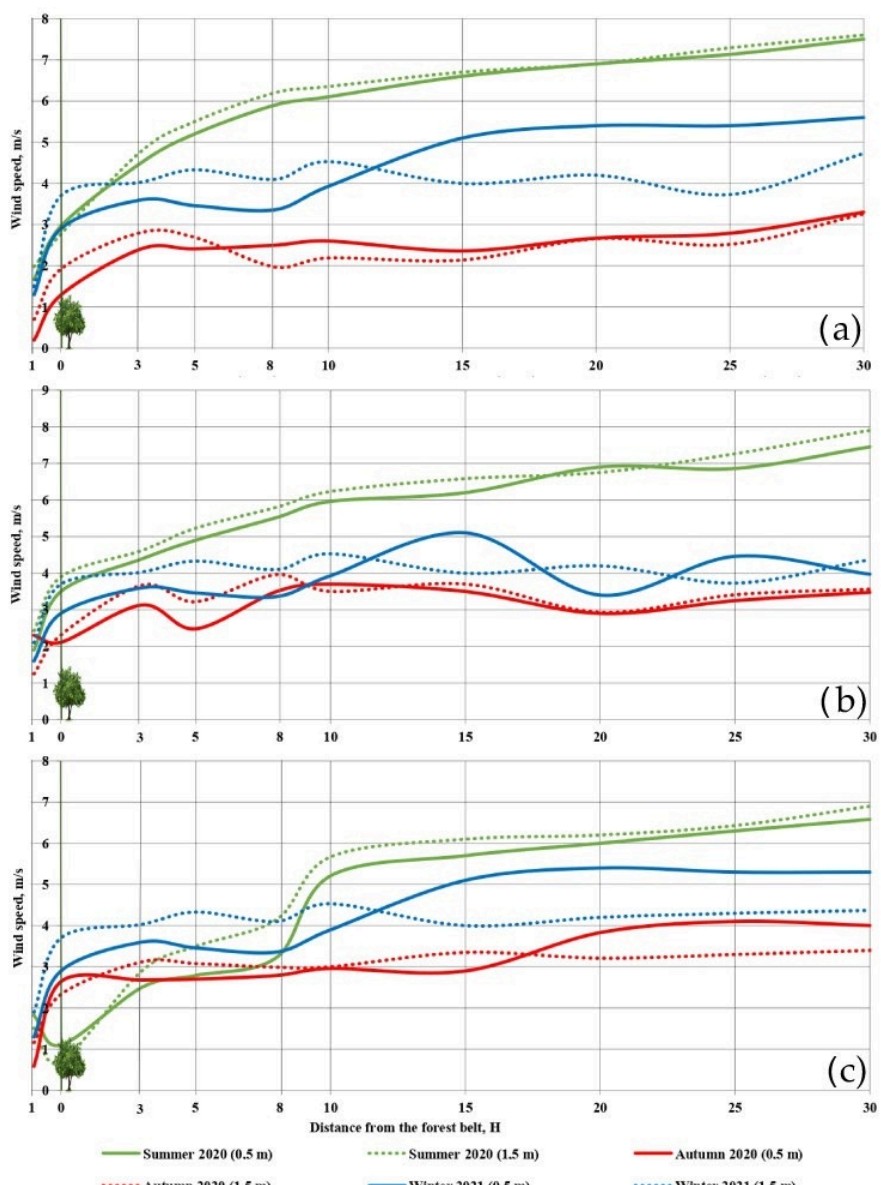

**Figure 9.** Wind speed in (**a**) the morning, (**b**) the afternoon, (**c**) and the evening.

In the afternoon in summer, there is also a "classical" impact of the forest belt at both studied altitudes, which can be traced up to 25–30H. Thus, the wind speed at a distance of 3H is 4.3 m/s and 4.6 m/s, respectively, and at a distance of 30H 7.5 m/s and 7.9 m/s, respectively. In autumn, the impact can be traced up to the distance of 8H at 0.5 m, and up to the distance of 10H at 1.5 m. In winter, the impact of the forest belt on the wind speed is not traced. Thus, the difference between the values of wind speed at the altitudes of 0.5 and 1.5 m and the distances of 3H and at 30H is 0.3 m/s.

In summer, in the evening time, the impact of the forest belt on the wind speed at the two studied altitudes is monitored up to a distance of 8–10H. The wind speed at the altitudes of 0.5 m and 1.5 m is 2.5 m/s and 2.9 m/s at 3H, respectively, and 6.6 m/s and 6.9 m/s at 30H, respectively. In autumn, at 0.5 m and 1.5 m, the impact of the forest belt on wind speed is on average up to a distance of 3H, and in winter—up to 8H.

## 4. Discussion

Agroforestry plantations have effects on various indicators, such as wind flow, snow distribution, the temperature of the surface air layer, and air humidity. Forest belts as complex, dynamic systems perform a flow-regulating function, protect against deflation and water erosion, and influence crop yields. In addition to biological functions, forest belts have socio-ecological, recreationalfunctions [50–56]. The impact of forest belts on the microclimate of agroforestry landscapes is undoubted. Field protection shelterbelt form their own special microclimate in the agrolandscapes of the surrounding area [31].

We noted that in modern studies many researchers [2,8,10,16,21] on climatic function of forest plantations focus on one or two climatic indicators (wind speed, air humidity, soil moisture, soil temperature, air temperature). Our research is comprehensive in nature. We evaluated wind speed, air temperature and humidity, and soil temperature and moisture, depending on the construction of the plantation. The age of the forest belt is of particular importance in assessing the effectiveness of the forest belt [57,58]. The age of stands determines the height of the stand, which in turn determines the range and degree of reclamation impact on the adjacent agrolandscape. The age of stands also determines the construction of the forest belt. After reaching a critical age, the tree species in the stand may lose their functions. The complexity of our research is due to covering different seasons. At the time of our microclimate measurements the forest belt was in different states: with and without leaves.

The effectiveness of the impact of forest belts on the microclimate of adjacent areas also depends on the construction of the forest belts [57,59]. The design of a forest belt is determined by two indicators: openness and wind permeability. Openwork is understood as the ratio of the area of openings in the longitudinal profile in the foliated state to its total area. Openwork strips act like lattice screens. The wind flow easily passes through the plantation, crushing and reducing its speed. Dense forest strips act as impermeable screens. The air stream flows around the plantation from above, and then drops down quite sharply at a distance of 3–5 heights, and at the same place, there is a mixing of surface air layers and the formation of wind erosion [60–62].

Table 1 summarizes the results of the positive effects of the forest belt on the elements of the microclimate.

**Table 1.** Summary table of the impact of the forest belt on microclimatic indicators in different seasons of the year.

| Microclimatic Parameter | Summer 2020 | Autumn 2020 | Winter 2021 |
|---|---|---|---|
| Wind speed, m/s | 25–30H | 15H | 15H |
| Air temperature °C | 10H | 10H | not impact |
| Soil temperature, °C | 15H | 10H | not impact |
| Air humidity, % | not impact | not impact | not impact |
| Soil moisture, % | 10–15H | 10H | not impact |

The research has shown that positive impact influence of forest belt on soil temperature in spring and autumn periods is traced up to 10–15H. In winter period, no impact is observed. The most distinct effect was noted during summer observations, so the average difference of temperatures in summer at 15H and in the forest belt was 28%. In autumn, the same difference was 15%. The positive impact on soil moisture in the autumn period is most clearly traced, so the average difference in soil moisture indicators near the forest belt and agrolandscape is 18%. In the summer period the similar difference is only 2%. That the direct effect of the forest belt on soil moisture in the summer season is fixed in the stand itself due to the shade and the formation of leaf litter, which prevents evaporation. In autumn, on the contrary, lower moisture values are recorded in the entire soil profile under

the forest belt as compared to the field. Impact of forest belt on air temperature reaches 20H in summer and autumn periods, but temperature difference is not great. The average temperature difference near the forest belt and at 20H is only 4%, and 1% in the summer period. According to the results we obtained, a significant effect on air humidity was not established. The only thing noted is that in the morning hours, the air humidity was higher in comparison with other periods. Effective reduction of wind speed occurs at 25–30H in the summer season, at 15H—in the autumn, at 15H—in the winter periods. The difference in wind speed near the forest belt and average temperature in the agrolandscape is 52% in summer, 40% in fall, and 30% in winter.

In general, the reasons for spatial differences in microclimate elements at different distances from the forest belt include the conditions of vertical air exchange, the state of soil cover and relief, time and season of the year, vegetative state of forest belts, their openness and height of the stand. Our studies have shown that wind speed depends on forest belt design and surface roughness (agrophonous or open surface), soil temperature depends on forest belt condition (presence or absence of phytomass), agrophonous, and microrelief in the field. Soil moisture dynamics in the zone of forest belt influence is determined by the presence of shade from the forest belt, wind speed, air temperature, season of the year, and local climatic conditions. Air humidity in the inter-belt fields depends on agrophonetics, air temperature, and soil moisture in the surface layer (up to 10 cm). The factors established by us are confirmed by other researchers [19,50].

The depression zone, which stands out at a distance of 5H, is the area along the forest belt, which experiences the greatest impact of the forest belt. In this zone, there are high values of moisture, both in individual layers and in the entire 50 cm thick layer, due to the accumulation of snow drifts in winter, due to the location of a number of shrubs. When growing crops in this zone, there is a competition for moisture between woody plants and cultivated plants, due to the exit of tree roots in the direction from the forest belt to the field. Such a zone is characteristic of the arid conditions of the chestnut soil zone [13,29]. As noted earlier, the question of variability of microclimatic parameters in the zone of the shelterbelt by seasons of the year and under different soil and climatic conditions remains relevant. In the absence of local comprehensive studies to assess the impact of protective forest plantations on the microclimate of the agrolandscape, our data can be used when comparing agrolandscapes of similar natural and climatic conditions. It should be noted that in arid low-forest regions, the forest function is partially performed by protective forest plantations. Their ameliorative impact due to increasing global climatic changes and drought is very high. We also believe that the restoration and maintenance of the optimal condition of protective forest plantations is an appropriate and important task to create optimal conditions for the cultivation of crops under conditions of insufficient moisture. The state of forest plantations determines their effective ameliorative functioning throughout their life span.

## 5. Conclusions

We studied the effect of forest belt parameters (structure, species composition, age) in different seasons on the complex of microclimatic factors (wind speed, soil temperature and humidity, air temperature and humidity). The interdependence between the distance from the forest belt and microclimatic indicators in agricultural landscapes of the Ilovlinsky district of the Volgograd region was established.

Our studies have shown that the positive effect of the openwork forest belt on the soil temperature in the spring and autumn periods can be traced to 10–15H in the first half of the day, while in the winter period, the impact is practically absent. Soil temperature and air temperature had the highest values in the second half of the day due to heating of surface air layer and surface of open soil as a result of high summer temperatures. Air humidity was inversely proportional to air temperature, and it was higher in the morning hours than in other time intervals. Wind speed decreased at 25–30H in the summer season, and 15H in the fall and winter seasons. The greatest wind effect was observed in the evening hours.

Differences in spatial microclimate indicators with respect to shelterbelt areas and unprotected areas of the field are primarily due to the design features of the forest belt and the time of day.

The results obtained are an attempt to assess the reclamation impact of the forest belt on microclimatic indicators in the current conditions of regional climate change, and taking into account the established relationships to make adjustments to the agricultural technologies used in the cultivation of crops in the interstrip space, as well as to develop ways to optimize the forest-melioration complexes. Climatic instability introduces uncertainty in the management of agrolandscapes at the local level, increasing the risks of failure to achieve the planned yields and incurring economic costs. Understanding the ameliorative role of forest belts in the formation of microclimate in the agrolandscape, it is necessary to ensure their proper condition for their long-term functioning, which in turn will reduce the aforementioned risks.

The results of the research correlate with the data of other authors, but they have regional features in terms of the zonal aspect, which highlights their novelty in the modern theory of forest reclamation of landscapes.

The obtained data on the impact of the forest belt on the complex of microclimatic indicators of the inter-belt space will be the basis for further research in terms of carbon deposition in the forest-meliorated area in the spatial and temporal context.

**Author Contributions:** Conceptualization and methodology, A.V.K.; formal analysis, Y.N.P.; investigation, A.V.K. and Y.N.P.; data curation, A.V.K. and Y.N.P.; writing—original draft preparation, A.V.K. and Y.N.P. All authors have read and agreed to the published version of the manuscript.

**Funding:** This study was funded of the Russian Ministry of Education and Science within the framework of State Assignment No 122020100312-0 "Theory and principles of formation of adaptive agroforestry-meliorative complexes of the dry steppe zone in the south of Russia in the context of climate change" Federal Scientific Center of Agroecology, Complex Melioration and Protective Afforestation, Russian Academy of Sciences.

**Data Availability Statement:** Not applicable.

**Acknowledgments:** We thank the Federal State Budgetary Scientific Institution "Federal Scientific Centre for Agroecology, Integrated Reclamation and Protective Afforestation of the Russian Academy of Sciences" for providing the technical equipment for the field research. We also thank K.N. Kulik, Academician of the Russian Academy of Sciences, for his valuable methodological advice in performing the research.

**Conflicts of Interest:** The authors declare no conflict of interest.

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
