# Peer review of "Impact of Field-Protective Forest Belts on the Microclimate of Agroforest Landscape in the Zone of Chestnut Soils of the Volgograd Region"

_forests, doi:10.3390/f13111892_

Round 1
Reviewer 1 Report (Previous Reviewer 1)
Line 31: delete redundant dot
Line 83-91: it would be helpful, if the timeframe for the for climate data is available (e.g. 1990-2010) and location of the weather station are specified
Line 99: as the rest of the manuscript omits the botanist's information, deleting the "L" from Triticum aestivum seems reasonable
Line 124: (beakers) ? redundant ?
Line 141: delete redundant blank
Line 148: please check if dash is correct
Line 176: sentence seems misplaced, would recommend to change
line 172 to: “… of 3H (equals 30m) from the forest belt….”
Line 182: check dash, please
Line 206: check dash, please
Line 210-212: move to discussion
Line 222-223: move to discussion
Line 242: Check redundant blank
Line 259: be consistent with times: 1 p.m.
Line 279-280: instead of difference I would recommend to explain whether the temperature is higher or lower
Line 358: Autumn, be consistent Line 384-390: why in bold?
Line 390: check heading, Conclusions
Line 403: autumn, be consistent Line 423: think CO2 is redundant
Author Response
Please see the attachment for the response to your comments. Thank you in advance.

Reviewer 2 Report (Previous Reviewer 2)
Shelterbelt plays an important role in improving the farmland microclimate environment. However, the observation data in this paper is sufficient, but the data analysis has not been deeply excavated. The data analysis should be further strengthened to improve the innovation of this paper. The main problems in this paper are as follows:
1. There is a lack of quantitative analysis in the abstract, and the percentage of meteorological elements falling or rising at different distances from the shelter forest should be put in the abstract.
2. The data table in the paper should be made into an intuitive diagram, for example, the data in the table 1, table 2 and table 3 should be transformed to diagrams, because bar charts or line charts are more intuitive than data tables and are more conducive to comparative analysis.
3. In the line 573 page 18, the serial number of 63. is redundant, please delete it.
4. The observation of microclimate elements by instruments settled at different distances from the shelter forest shall indicate that the instruments are set in the downwind direction rather than the upwind direction of the shelter forest. What is the perennial prevailing wind direction in the study area?
5. Why is there no data observation of meteorological elements inside and outside the shelter forest in spring in the paper?
6. In the line 113-115 page 8, don't specify who carried out the experiment, just make it clear where it was done, and please delete the sentences.
7. In the line 390 page 14, the title of 5. Conclusions should be on the next line, the words in the part of lines from 384 to 390 in page 14 shouldn’t be bold font. The discussion part is divided into too many paragraphs, which seems to have unclear logical relationship; two or three paragraphs are enough for the discussion part.
8. In the line 176 page 6, the sentence of “Height (Н) of the forest belt 10 meters.” is redundant, please delete it.
9. The analysis of meteorological data in this paper is very vague. It should be that the monitoring of all meteorological elements is the three-month 90 day of average meteorological elements in the morning, noon and evening of each season, while the data analysis does not say how to calculate the average value of the daily data in three seasons.
10. The discussion part of the paper lacks the comparative analysis of similar research conclusions with others, lacks certain innovation, and has less analysis of the reasons for the differences between research conclusions and others.
Author Response
Please see the attachment for the response to your comments. Thank you in advance.

Reviewer 3 Report (New Reviewer)
The study brings results on the research of the effect of forest belts on microclimate conditions. Such research is important to support establishing and optimizing such practices in agriculture landscape and The biggest deficiency of this study is very low number of repetitions both in space (one site) and in time (three times per year). It is difficult to make the conclusions from one day measurement per season. I am also not convinced that interpolation (or curve fit) of soil parameters (e.g. Fig 5) between FB and 3H is correct to display the changes in the parameters with distance. I am not also sure that any courses in e.g. soil temperature (Fig. 5) between 3H and 30H are systematic, but more like measurement deviations due to low replications. On the base of this, I must say that such data are not enough for highly raked Journal as Forests, and I cannot support it for publication. Bellow, there are a few specific comments which might be helpful for the authors.
Specific comments
L36: Please, use only global change (it is usually used global change or climate change)
L121: It seems to me strange that the thermoanemometer can measure soil temperature. It probably needs different sensors for air and soil. Should be more detailed
L113-114: when exactly (dates)? One day per season? How did you choose the days?
L132-133: so you have any parameter of deviation of the averages (e.g. SD)
L152: the listing of the measured parameters is a repetition of L120-125. It would be also more appropriate to keep the information on the measurements together
L154: 3H is missing in Fig. 1
L154 and 157: please explain why the distances for air and soil parameters are not same
L156: please specify depth the soil parameters were measured
Figure 5: please, what is number 1 on the x-axis on the left from 0? I tis not in the measurement scheme in Fig. 4
The statements what new or innovative is brought by your study (L318 and L326) should be rather in Introduction.
L344-358 are just a repletion of the results
Author Response
Please see the attachment for the response to your comments. Thank you in advance.

Round 2
Reviewer 3 Report (New Reviewer)
The authors considered and answered all comments and improved the manuscript. I have only a few minor comments.
Thank you for adding the H1 measurement point to Figure 4. It should be, however, added also to the text L184-185
Figures 2, 3, 5, and 9: For keeping consistency with new figures, please use a), b) c) instead of I, II, III for individual parts.
If you can, please move the y-axis label to the left axis where are the numbers in Figures 7 and 8
Author Response
Thank you for your comments. Please see the attachment for the response.

This manuscript is a resubmission of an earlier submission. The following is a list of the peer review reports and author responses from that submission.
Round 1
Reviewer 1 Report
Abstract:
Line 10-23: A more detailed description of the impact of the forest belt regarding the measured microclimatic parameters on the adjacent cropland, expressed relative to unsheltered conditions or at least their impact range, would greatly enhance the quality of this abstract. At the same time, the present study results should be briefly elicited in the context of the research question and thus its relevance to the agroforestry landscape described.
Introduction:
Line 42: H instead of N
Materials and Methods:
Figure 1D: scale is not legible
Line 61-62: long-term average air temperature is missing, possibly semi arid climate ?
Line 68 - 69: what means agro-alkali soils ? clay (alkali) soils with agriculture ? should be better explained or referenced if necessary
Line 70: add Triticum aestivum L.
Line 72-73: coordiantes could be moved to Line 57.
Line 76-89: please add reference of the soil classification system used
Line 98-104: technical information of sensors/measuring equipment used are missing
Line 109: plant density ?
Line 112-114: reference to the tree quality approach is missing
Line 118: maybe porosity is meant ?!
Line 120: add "(Exelis Visual Information Solutions, Boulder, Colorado)".
Line 127: reference missing
Line 128-136: information regarding data preparation and soil sampling (depth) and method of soil moisture determination are missing
Results and discussion
Line 160: 1H instead of 1N, I would also recommend to add information in metres, at least for the first time introducing this specification in this chapter
Line 147-172: It is not entirely clear if these seasonal data is taken from a single day or if they are averages for the whole season ... the same applies for the remaining study parameters
Line 185: since relative humidity will also be discussed later, I would recommend consistently referring to soil moisture.
Line 190: en dash not appropriate here, please check
Line 194: statistical evaluation ? p-value ?
Line 208: see line 185
Figure 7: heading (soil moisture in summer/autumn) are redundant
Line 228-229: cant see that, in forest belt the difference is 1.1°C in 25H its 0.5°C
Line 239/242: please check wheather en dash is appropriate here
Line 243/244: cant see that, air temperature at 3H in both altidudes are colder than in 30H
Line 282: en dash not appropriate here, please check
Line 288: H instead of N
Table 3: Information regarding the evaluation of the strength of influence in the material and methods chapter would have been helpful for the comprehensibility of the evaluation made here. For example, why is there no mention of the influence of the tree belt on relative air humidity in 1H? In summer, for example, here it is considerably lower ( ~20%) than in the other positions in the field. On the other hand, with the data available, it is not quite clear to me why the air temperature should be affected by the tree belt up to a disctance of 20H in summer/autumn ?!
Line 146-304: Generally main results discussed with references are missing here
Conclusions
Line 305-327: Should be revised in view of the following or at least some of the follwing questions/keywords:
uncertainties ? recommendations ? impact on agricultural landscape/crop prodution ? future research questions ?
Author Response
Response to Reviewer 1 Comments
Point 1: Line 10-23: A more detailed description of the impact of the forest belt regarding the measured microclimatic parameters on the adjacent cropland, expressed relative to unsheltered conditions or at least their impact range, would greatly enhance the quality of this abstract. At the same time, the present study results should be briefly elicited in the context of the research question and thus its relevance to the agroforestry landscape described.
Response 1: We have re-edited the abstract according to your recommendations
Point 2: Line 42: H instead of N
Response 2: Сomment was corrected
Point 3: Figure 1D: scale is not legible
Response 3: Тhe picture was corrected
Point 4: Line 61-62: long-term average air temperature is missing, possibly semi arid climate ?
Response 4: Introduced information on air temperature for the study area
Point 5: Line 68 - 69: what means agro-alkali soils ? clay (alkali) soils with agriculture ? should be better explained or referenced if necessary
Response 5: The translation mistake has been corrected. I meant chestnut solonetz and chestnut agrosolonetz soils
Point 6: Line 70: add Triticum aestivum L.
Response 6: Сomment was corrected
Point 7: Line 72-73: coordiantes could be moved to Line 57
Response 7: Сomment was corrected
Point 8: Line 76-89: please add reference of the soil classification system used
Response 8: We decided to delete the information about the morphological description of the transect under the forest belt along with the figure, as this information had no meaningful meaning.
Point 9: Line 98-104: technical information of sensors/measuring equipment used are missing
Response 9: Technical information about the sensors and measuring equipment used has been added
Point 10: Line 109: plant density ?
Response 10: Сomment was corrected
Point 11: Line 112-114: reference to the tree quality approach is missing
Response 11: Reference added
Point 12: Line 118: maybe porosity is meant ?!
Response 12: Сomment was corrected. It means to the degree of gaps in the intercrown space of the vertical profile of the plantation, this is called openwork.
Point 13: Line 120: add "(Exelis Visual Information Solutions, Boulder, Colorado)".
Response 13: Сomment was corrected
Point 14: Line 127: reference missing
Response 14: Reference added
Point 15: Line 128-136: information regarding data preparation and soil sampling (depth) and method of soil moisture determination are missing
Response 15: This information is contained in lines 224-229 in the corrected version of the manuscript
Point 16: Line 160: 1H instead of 1N, I would also recommend to add information in metres, at least for the first time introducing this specification in this chapter
Response 16: Сomment was corrected
Point 17: Line 147-172: It is not entirely clear if these seasonal data is taken from a single day or if they are averages for the whole season ... the same applies for the remaining study parameters
Response 17: Measurements were taken once in each season summer, autumn, winter, at 3 time intervals during the day at 10 a.m., 1 p.m., 4 p.m., at 2 heights of 0.5 m and 1 m
Point 18: Line 185: since relative humidity will also be discussed later, I would recommend consistently referring to soil moisture.
Response 18: Сomment was corrected
Point 19: Line 190: en dash not appropriate here, please check
Response 19: Сomment was corrected
Point 20: Line 194: statistical evaluation ? p-value ?
Response 20: Statistical processing of the obtained data was performed at the 95% significance level, but was not presented in the manuscript
Point 21: Line 208: see line 185
Response 21: Сomment was corrected
Point 22: Figure 7: heading (soil moisture in summer/autumn) are redundant
Response 22: Сomment was corrected
Point 23: Line 228-229: cant see that, in forest belt the difference is 1.1°C in 25H its 0.5°C
Response 23: Сomment was corrected. You're right. It was a technical mistake.
Point 24: Line 239-242: please check wheather en dash is appropriate here
Response 24: Сomment was corrected
Point 25: Line 243-244: cant see that, air temperature at 3H in both altidudes are colder than in 30H
Response 25: Corrections have been made to this sentence. What was meant was that in the forest belt itself, the temperature is higher than in the 3H.
Point 26: Line 282: en dash not appropriate here, please check
Response 26: Сomment was corrected
Point 27: Line 288: H instead of N
Response 27: Сomment was corrected
Point 28: Table 3: Information regarding the evaluation of the strength of influence in the material and methods chapter would have been helpful for the comprehensibility of the evaluation made here. For example, why is there no mention of the influence of the tree belt on relative air humidity in 1H? In summer, for example, here it is considerably lower ( ~20%) than in the other positions in the field. On the other hand, with the data available, it is not quite clear to me why the air temperature should be affected by the tree belt up to a disctance of 20H in summer/autumn ?!
Response 28: A Discussion section has been added to the manuscript.. This section presents an analysis of the results. 1H is a zone on the windward side of the forest belt, in which there is an increase in wind flow, due to the cooling effect there is a decrease in air humidity. The decrease in air temperature during the summer/autumn season is affected by the forest belt by up to 10H. 20H is an mistake.
Point 29: Line 146-304: Generally main results discussed with references are missing here
Response 29: The text is partially edited and moved to the Discussion section along with the references
Point 30: Line 305-327: Should be revised in view of the following or at least some of the follwing questions/keywords: uncertainties ? recommendations ? impact on agricultural landscape/crop prodution ? future research questions ?
Response 30: The conclusions in the manuscript have been revised as recommended
We want to thank you very much for the detailed review of our manuscript and the constructive comments that have certainly improved our manuscript. We have tried to take all your comments into account.
Best regards, authors

Reviewer 2 Report
1.The main research conclusions are not stated in the abstract in the paper.
2.The introduction lacks the relevant research progress at home and abroad, and introduces the physical and chemical properties of farmland soil too much in the research materials and methods part, the soil properties have no close relationship with the microclimate in shelter belt.
3.The data table in the paper should be made into an intuitive diagram, for example,the data in the table 1 and table 2 should be transformed to diagrams.
4.The paper lacks discussion part, and the innovation of the paper is insufficient.
5.Is the observation data of microclimate observed three times every day in three seasons or one day in each season? The data source was not clearly stated in the paper.
6.The order of references cited should be from small to large number, while the order of references cited in this paper is chaotic.
7.In some places, the letter "H" is replaced by the letter "N", for example, in the line 88 page 13, the letter of N behind the letter of 10 should be the letter of N.
8.The reasons behind the spatial differences of microclimate elements at different distances of shelter forests are rarely explained and analyzed, and the analysis should be strengthened.
9.For accuracy, all data in the paper should have at least one decimal point instead of an integer.
10.The experimental data are not fully mined and analyzed, so the conclusion of the paper is not very innovative and theoretical value.
Author Response
Response to Reviewer 2 Comments
Point 1: The main research conclusions are not stated in the abstract in the paper.
Response 1: We have re-edited the abstract according to your recommendations
Point 2: The introduction lacks the relevant research progress at home and abroad, and introduces the physical and chemical properties of farmland soil too much in the research materials and methods part, the soil properties have no close relationship with the microclimate in shelter belt.
Response 2: We have rewritten the Introduction and added the progress of research at home and abroad on the problem under study, in accordance with your recommendation
Point 3: The data table in the paper should be made into an intuitive diagram, for example,the data in the table 1 and table 2 should be transformed to diagrams.
Response 3: We decided not to convert the data in Tables 1 and 2 into a graphical part, since the manuscript would be overloaded with graphical material, where it is not always possible to accurately trace quantitative changes in the data. Please understand us correctly.
Point 4: The paper lacks discussion part, and the innovation of the paper is insufficient.
Response 4: A discussion section has been added to the manuscript in accordance with your recommendation, indicating the novelty of our work, as in the introduction.
Point 5: Is the observation data of microclimate observed three times every day in three seasons or one day in each season? The data source was not clearly stated in the paper.
Response 5: Measurements were taken once in each season summer, autumn, winter, at 3 time intervals during the day at 10 a.m., 1 p.m., 4 p.m., at 2 heights of 0.5 m and 1 m
Point 6: The order of references cited should be from small to large number, while the order of references cited in this paper is chaotic.
Response 6: The citation order is orderly, and references are given as they are used
Point 7: In some places, the letter "H" is replaced by the letter "N", for example, in the line 88 page 13, the letter of N behind the letter of 10 should be the letter of N.
Response 7: Сomment was corrected
Point 8: The reasons behind the spatial differences of microclimate elements at different distances of shelter forests are rarely explained and analyzed, and the analysis should be strengthened.
Response 8: The reasons for the spatial differences of microclimate elements at different distances from the forest belt are presented in the Discussion part, a brief analysis of them is also conducted
Point 9: For accuracy, all data in the paper should have at least one decimal point instead of an integer.
Response 9: Сomment was corrected
Point 10: The experimental data are not fully mined and analyzed, so the conclusion of the paper is not very innovative and theoretical value.
Response 10: The conclusions of the article are revised according to the comment. The data are analyzed in the discussion part.
We want to thank you very much for your review of our manuscript and your constructive comments, which certainly improved our manuscript. We have tried to take most of your comments into account.
Best regards, authors
